# Personalized Collaborative Fine-Tuning for On-Device Large Language Models

**Nicolas Wagner, Dongyang Fan, Martin Jaggi**
EPFL, Switzerland
nwepfl20@gmail.com, {dongyang.fan, martin.jaggi}@epfl.ch

## Abstract

We explore on-device collaborative fine-tuning of large language models under limited local data availability. We introduce three distinct dynamic collaborator selection schemes, allowing trust-weighted personalized update aggregation: model-similarity-based, prediction-similarity-based and validation-performance-based. To minimize communication overhead, we integrate Low-Rank Adaptation (LoRA) and only exchange LoRA model updates. Our protocols, driven by prediction and performance metrics, surpass both FedAvg and local fine-tuning methods, which is particularly evident in realistic distributed scenarios with more diverse local data distributions. The results underscore the effectiveness of our approach in addressing heterogeneity and scarcity of the local datasets.

## 1 Introduction

Recently, there has been an unprecedented surge in the popularity of Large Language Models (LLMs), driven by their versatile capability to solve a wide range of tasks and serve as general-purpose models (Touvron et al., 2023; OpenAI et al., 2024; Jiang et al., 2024). Their downstream performances can be further enhanced by fine-tuning, which is typically conducted on fewer data and independently by different users, rather than in a centralized way. The scarcity of local data often renders local fine-tuning ineffective, necessitating collaboration. However, in many cases, end users usually have privacy concerns over their local data, such as patient records for hospitals (Li et al., 2023) and typing history for mobile phone users (Hard et al., 2018). Naturally, one might wonder *how we could enable users to collaborate to still obtain better models in the presence of small and privacy-sensitive local data.*

The possibility of on-device fine-tuning of LLMs has been facilitated by parameter-efficient techniques such as prompt tuning (Lester et al., 2021), adapters (Houlsby et al., 2019) and Low-Rank Adaptation (LoRA, Hu et al. (2022)). LoRA has gained widespread popularity for fine-tuning large language models since its introduction, which approximates model updates through the multiplication of low-rank matrices. LoRA modules align with our collaboration initiative, as they allow for a significant reduction in communication overhead. A naive approach to leveraging local data from other users would be to average LoRA weights, which is the popular FedAvg algorithm proposed by McMahan et al. (2017). However, the resulting one global model solution might not fit all users' data distributions, especially when there is more heterogeneity present. As a result, we aim to develop personalized collaboration strategies for each user, which should surpass the performance of local fine-tuning and naive averaging approaches.

In pursuit of this objective, we introduce several collaborator selection protocols tailored for optimal LoRA weight averaging across diverse user profiles. While similar concepts have been extensively investigated within collaborative learning communities (Zhang et al., 2021; Sui et al., 2022; Fan et al., 2024), these approaches are only designed for supervised classification tasks and thus are not directly applicable to Large Language Models (LLMs). Nevertheless, we investigate how we can adapt and apply analogous principles to *determine optimal aggregation weights for LoRA matrices on a per-user basis.*

---

Our code is available at https://github.com/epfml/personalized-collaborative-llms

Following some computational social science literatures (Urena et al., 2019; Zhang et al., 2022), we term the aggregation weight matrix "trust", as if we view clients as members of a social group, their exchanged information can thus be regarded as opinions. Similarity measures can be useful for assessing trust between users, as users are more likely to consider and trust others with similar opinions. Such similarity measures are core to our proposed protocols, which we will discuss in Section 3.1.

Our contributions can be summarized in the following aspects:

- We are the first to tackle *realistic* data heterogeneity among users in personalized collaborative language modeling, encompassing diverse topic distributions and varying language usage.
- We demonstrate that collaboration can be leveraged to improve personalization performance in language modeling and show that predictions are more effective than model parameters for identifying collaborators in the language domain.
- Our approach is well-suited for on-device fine-tuning, effectively mitigating challenges related to data scarcity and resource limitations.

## 2 Related Work

We focus on a *peer-to-peer decentralized learning* setting, where the existence of a central server is not assumed. Instead, the end users conduct peer-to-peer communication using decentralized schemes such as gossip averaging to aggregate local information across agents. To enable a personalized weighted aggregation, several recent works have proposed data-dependent communication protocols based on task similarities and node qualities. Notably, Zhang et al. (2021) derives a first-order approximation for optimal aggregation weights $w_{ij}^\star$ for collaboration, which happens to be proportional to how well client $j$'s model generalizes on client $i$'s data. Li et al. (2022) directly optimizes the mixing weights by minimizing the local validation loss per node. Sui et al. (2022) uses the E-step of the EM algorithm to gauge the significance of other agents for a specific agent $i$, achieved through assessing the accuracy of these agents' models on the local data of agent $i$. Fan et al. (2024) adopt a different approach by comparing prediction similarity on a publicly available dataset, which largely reduces communication overhead and allows for model heterogeneity. These approaches have been showing promising performances in supervised deep learning experiments. For language models, the data heterogeneity across users is not well studied, and the effectiveness of collaborator selection methods has not been explored for LLMs, where self-supervised training is performed.

Recently, groups of researchers have been investigating the intersection of Federated Learning and Large Language Model training. Due to the substantial number of model parameters, these efforts primarily focus on post-training stages, often incorporating parameter-efficient techniques (Fan et al., 2023; Ye et al., 2024; Flower.ai, 2024; Flare, 2024). Federated pre-training was studied in (Sani et al., 2024). Zhang et al. (2023) employ FedAvg for collaborative instruction tuning, where they demonstrate superior performance in task generalization compared to local instruction tuning. Che et al. (2023) design an adaptive optimization method for collaborative prompt tuning, where a scoring method is applied to measure the importance of each layer and only prompt parameters of more important layers are exchanged during the tuning process, ensuring communication efficiency. Cho et al. (2024) tackles the challenge brought up by different LoRA ranks on heterogeneous devices. By performing rank self-pruning locally and sparsity-weighted aggregation at the server, improved convergence speed and final performance can be achieved. In contrast to these methods, we aim at providing personalized models to each user, instead of a global solution.

## 3 Method

To start with, we first introduce how LoRA works. For a pre-trained model parameterized with $\mathbf{\Phi} \in \mathbb{R}^{m \times n}$, the model updates during fine-tuning stage can be approximated by

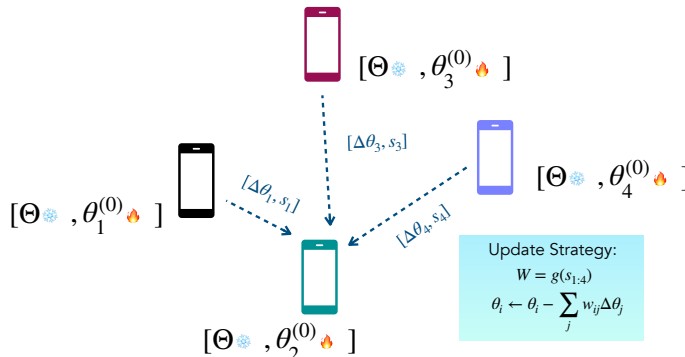

Figure 1: Diagram of our protocol. $\theta_i$ and $\Delta\theta_i$ represent LoRA parameters and LoRA model updates respectively. $s_i$ denotes messages to send beside $\Delta\theta_i$, which represents either $\theta_i$ or $f_{\theta_i}(X_S)$ depending on the protocol (see Table 1). $g(\cdot)$ is our proposed trust calculation approach as detailed in Section 3.2.

multiplication of two low-rank matrices $A \in \mathbb{R}^{m \times r}$, $B \in \mathbb{R}^{r \times n}$

$$\Delta\Phi \approx AB, \qquad \text{where} \qquad r \ll \min(m, n) \tag{1}$$

We work in a setting where all users have the same LoRA rank $r$, and aim to arrive at their personalized $\Delta\Phi_i$. During the fine-tuning phase, only LoRA parameters will be updated, and the original weights of the pre-trained model will remain frozen. Therefore, it suffices to communicate LoRA models, instead of the big chunk of whole parameters.

### 3.1 Our Protocol

We stick to the standard approach of sharing and aggregating model updates. Each user, through their unique data distribution, contributes to the collective learning process by sharing local model updates. Central to our approach is the so-called *trust matrix*, which we define as the gossip aggregation matrix to guide each user's aggregation of model updates. The calculation of trust matrix $W$ will be detailed in Section 3.2. We use $W_{i\cdot}$ to denote the $i$th row of $W$. A simplified diagram of our protocol is shown in Figure 1.

Our proposed protocol is described in Algorithm 1. To summarize, the protocol is constituted with three phases: 1) each user $i \in [N]$ computes their LoRA model update $\Delta\theta_i$ with respect to their local data (and if required, predictions $f_{\theta_i}(X_S)$ with respect to the shared dataset $X_S$); 2) each user communicates $[\theta_i, \Delta\theta_i]$ or $[f_{\theta_i}(X_S), \Delta\theta_i]$ to all other users, depending on the trust calculation strategy; 3) each user $i \in [N]$ calculates their unique trust weights $W_{i\cdot}$ locally in parallel and updates their LoRA parameters using trust-weighted received model updates. It is noteworthy that different means of trust weight calculation require different information to be shared across the users, thus inducing distinct computation complexities and communication overheads. We will discuss those aspects in Section 3.3.

### 3.2 Trust-based Collaborator Selection

In this section, we explore different ways of building trust – gossip aggregation graph across users. Essentially, the weight of edge $(i, j)$ should denote to what extent can user $j$'s model help to facilitate user $i$'s learning progress.

#### 3.2.1 Model-similarity-based

We employ pairwise LoRA parameter similarity to compute trust scores among users. The underlying idea is that users whose model parameters exhibit closer alignment likely possess data distributions that are more similar, thereby rendering their contributions more pertinent and advantageous to one another. This methodology shares a resemblance with (Li et al.,

---

**Algorithm 1** Our proposed protocol

---

**Require:** Number of communication rounds T, pre-trained model $\boldsymbol{\Theta}$, initialized LoRA models $\boldsymbol{\theta}_i^{(0)}$, local dataset $\mathbf{X}_i$, shared dataset $\mathbf{X}_S$ (required for strategy 3.2.3).

  **for** $t = 1, ..., T$ **do**
    **in parallel** for each user $i \in [N]$ **do**
        1. Compute personal update $\Delta\boldsymbol{\theta}_i^{(t-1)}$ on $\boldsymbol{\theta}_i^{(t-1)}$ with respect to local data $\mathbf{X}_i$
        2. Broadcast each $\Delta\boldsymbol{\theta}_i^{(t-1)}$ to all others agents
        3. Compute collaboration weight matrix $\mathbf{W}$,
                    based on trust-based strategy 3.2.1, 3.2.2 or 3.2.3
        Compute trust-weighed personalized model update:

$$\vartheta_i^{(t)} = \sum_{j\in[N]} w_{ij}\Delta\boldsymbol{\theta}_j^{(t-1)}$$

        Update personal LoRA parameters, with learning rate $\eta$

$$\boldsymbol{\theta}_i^{(t)} = \boldsymbol{\theta}_i^{(t-1)} - \eta\vartheta_i^{(t)}$$

  **end for**

---

2022), where the mixing weight is derived from the inner product between representations of different users' models via an encoder model. However, our approach involves directly measuring the similarity between LoRA models without needing an encoding module.

It appears that in addition to LoRA model updates, LoRA models also need to be shared to compute the trust matrix. However, this additional communication can be efficiently circumvented, as the updates to LoRA models are already exchanged, and the models themselves can be readily derived by aggregating these updates in each iteration. The collaboration trust matrix $\mathbf{W}$ is computed as follows:

$$\tilde{w}_{ij}^t = \frac{\langle\boldsymbol{\theta}_i^{(t-1)}, \boldsymbol{\theta}_j^{(t-1)}\rangle}{\|\boldsymbol{\theta}_i^{(t-1)}\|\|\boldsymbol{\theta}_j^{(t-1)}\|}, \qquad W^t = \text{SoftMax}(\widetilde{\boldsymbol{W}}^t, \dim = 1) \tag{2}$$

### 3.2.2 Validation-performance-based

Drawing inspiration from Zhang et al. (2021), we evaluate the pairwise trust $w_{ij}$ based on how well user $j$'s model performs on the validation sets of user $i$. This method requires extra validation sets within each user, on which the performance assessment directly reflects the relevance and potential benefit of one user's model to another. The aim is to guide the aggregation process by favoring models that demonstrate compatible performance levels, thereby enhancing the overall effectiveness of collaborative learning. The calculation is denoted as:

$$\tilde{w}_{ij}^t = \mathcal{L}(f_{\boldsymbol{\theta}_j^{t-1}}(\boldsymbol{X}_i^{\text{val}}), \boldsymbol{X}_i^{\text{val}}), \qquad W^t = \text{SoftMax}(-\widetilde{\boldsymbol{W}}^t, \dim = 1) \tag{3}$$

where $f_{\boldsymbol{\theta}_j^{t-1}}(\boldsymbol{X}_i^{\text{val}})$ denotes a forward pass of user $j$'s model on user $i$'s local validation set, and $\mathcal{L}$ is the cross entropy loss for next token prediction.

### 3.2.3 Prediction-similarity-based

In the prediction-similarity-based approach, following the trust weighing strategy from Fan et al. (2024), we derive aggregation weights by comparing the model's predictions on a shared dataset, in this case, logits across the vocabulary. A public dataset $\boldsymbol{X}_S$ will be added to the input of Alg. 1, and additionally communicating those predictions between users is

necessary, increasing communication cost. This method assesses how closely the predictions of different models align, prioritizing contributions from users whose models exhibit similar predictive behaviors. The trust is calculated based on $l_1$ distance between logits:

$$\tilde{w}_{ij}^t = |f_{\boldsymbol{\theta}_i^{t-1}}(\boldsymbol{X}_S) - f_{\boldsymbol{\theta}_j^{t-1}}(\boldsymbol{X}_S)|, \qquad \boldsymbol{W}^t = \text{SoftMax}(-\widetilde{\boldsymbol{W}}^t, \dim = 1) \qquad (4)$$

### 3.3 Remarks

Table 1 provides an overview of the communication and computation costs associated with the three proposed strategies.

**On communication costs.** In terms of communication, both strategies 1 and 2 need to share LoRA models ($\boldsymbol{\theta}_i$) on top of LoRA model updates ($\Delta\boldsymbol{\theta}_i$). Strategy 3 requires sharing logits ($f_{\boldsymbol{\theta}_i}(\boldsymbol{X}_S)$) across the public dataset, rather than LoRA models. Although the logits encompass the entire vocabulary, it is possible to heavily reduce its memory cost by artificially transforming it into a sparse matrix. The computation of trust through prediction similarity relies on the $l_1$ distance, and most entries of logits tend to be very close to zero, making it feasible to only keep the top-k logits values. This approach greatly cuts down on communication costs when sharing logits while still maintaining desired performance levels. For a detailed comparison of the measured communication costs, please refer to Appendix A.1. Nonetheless, as the communication of $\Delta\boldsymbol{\theta}_i$s remains unavoidable and constitutes the primary communication cost, there is not much variation in communication costs across all three strategies.

**On extra computation costs.** Regarding additional computation costs to obtain $W$, Strategy 1 involves only calculating cosine similarities between LoRA models in addition to existing processes, the cost of which is in the order of the number of LoRA parameters. However, Strategies 2 and 3 necessitate multiple forward passes to generate predictions either on local validation sets or the publicly shared dataset. A forward pass is significantly more resource-intensive compared to cosine similarity calculations due to the model's size. In Strategy 2, evaluating all models' performance on each local validation set requires $N^2$ model inferences. In Strategy 3, each user needs extra forward passes to obtain predictions on $\boldsymbol{X}_S$, resulting in $N$ times the model inference. Since $\boldsymbol{X}_S$ and $\boldsymbol{X}_i^{\text{val}}$ are similar in size, the number of forward passes per inference doesn't vary much. Thus, Strategy 2 requires roughly $N$ times more extra computation cost than Strategy 3.

| Strategy | Communicated Elements | Extra Computation Costs | |
|---|---|---|---|
| 1: Model-sim | $\{\boldsymbol{\theta}_i, \Delta\boldsymbol{\theta}_i\}_{i=1}^N$ | $\mathcal{O}(Ldk)$ | $\star$ |
| 2: Validation | $\{\boldsymbol{\theta}_i, \Delta\boldsymbol{\theta}_i\}_{i=1}^N$ | $\mathcal{O}(Ln^2d + Ld^2n) \cdot N^2$ | $\star\star\star$ |
| 3: Predictions | $\{f_{\boldsymbol{\theta}_i}(\boldsymbol{X}_S), \Delta\boldsymbol{\theta}_i\}_{i=1}^N$ | $\mathcal{O}(Ln^2d + Ld^2n) \cdot N$ | $\star\star$ |

Table 1: A comparison of communication and computation complexity across the strategies. $L$ denotes the number of layers, $d$ denotes the embedding size, $n$ denotes context length and $k$ denotes LoRA rank. $N$ denotes the number of users in the system. It is clear Strategy 2 adds the most extra computational cost, followed by Strategy 3. Strategy 1 requires the least additional amount of computation.

## 4 Experiments

### 4.1 Setup

#### 4.1.1 Experimental details

Our experimental setup is structured as follows: Each user is configured with a uniform model architecture, to ensure model updates can be shared and aggregated. We chose

relatively small LLMs constrained by the limited computing resources in academia. For each user, we equip them with a GPT2[1] base model of 124 million parameters. It has 12 layers, 12 attention heads, embedding size 768 and vocabulary size 50257.

LoRA modules were applied on top of both Causal Self-Attention (SA) and MLP layers, constituting 0.47% of full parameters. The reason why we chose the two blocks is a trade-off between the model performance and communication overhead, as shown in Figure 2. Although incorporating LoRA modules onto Self Attention layers results in the most significant reduction in test perplexity per trainable parameter, integrating LoRA modules into MLP layers can contribute to further decreasing test perplexity while maintaining a reasonable number of trainable parameters.

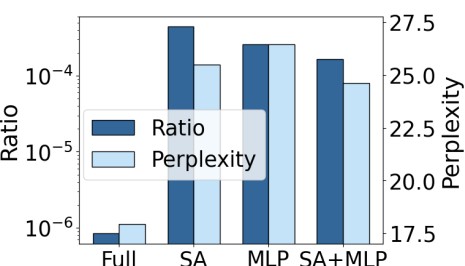

Figure 2: Ablation study of whether to add LoRA modules compared to full fine-tuning. Left bars correspond to *Ratio*, which denotes the decrease in test perplexity compared to the pre-trained model per trainable parameter (higher is better); right bars correspond to *perplexity*, which denotes test performance after fine-tuning (lower is better).

The experiments were conducted, for each client, with a learning rate of 0.002, a batch size of 50 with 4 accumulation steps, a context length of 512, and a total of 500 iterations. Every user performs local training for 100 iterations before communication, so that the local models can start to diverge due to the variation of local data distributions. Afterward, communication and aggregation are done every 25 iterations. As we target a limited data regime, the number of iterations ensures that over 50 epochs of training are conducted in most datasets. Throughout all experiments, we use the same LoRA configurations: a rank of 4, an $\alpha$ value of 32.0, and a dropout rate of 0.1. The selection of hyperparameters is determined through a sweep of various values while considering the limitations posed by our computing resources, as presented in Appendix A.2.

### 4.1.2 Datasets

In vision tasks, realistic data heterogeneity arises from diverse domain distributions across clients and variations in image quality resulting from the use of different devices for data acquisition (Ogier du Terrail et al., 2022). However, within the domain of written language, realistic data heterogeneity frequently arises from variations between sources. Owners of such sources can differ in writing styles, vocabulary choices, grammar structures, and topic distribution. Our experiments are designed to examine diverse topic distributions and language usages exhibited by different users.

We focus on on-device fine-tuning scenarios, where the amount of available local training data may be limited. The diverse local data distribution originates from user-specific characteristics. For instance, users often possess unique topic preferences concerning news reading, and mobile phone users often type in different languages, prompting smart keyboards to enhance next-word prediction across different language blends. Following this, we investigate two levels of data heterogeneity: 1) *Low heterogeneity*, where each user is assigned two data categories from the entire set, and a $(3/4, 1/4)$ mixture of data of these categories is allocated to each user. 2) *High heterogeneity*, where each user is exclusively assigned to one data category.

The descriptions of the three dataset partitionings utilized in our experiments are as follows:

1. AG News: News articles in four categories: "World", "Sports", "Business" and "Sci/Tech". (Zhang et al., 2015)
2. Multilingual Wikipedia: Wikipedia texts in three languages (categories): French, Italian, and German. (Wikimedia, 2022)

---

[1]https://github.com/karpathy/nanoGPT

| Datasets | # users | # Categories | # Training tokens | # Test tokens |
|----------|---------|--------------|-------------------|---------------|
| AG News | 4/8 | 4 | $\sim 1'500'000/\sim 750'000$ | $\sim 75'000/\sim 50'000$ |
| Multi-Wiki | 9 | 3 | 840'000 | 160'000 |
| Codes-Wiki | 8 | 2 | 840'000 | 160'000 |

Table 2: Dataset configurations used in the experiments. Reference data $X_S$ is slightly larger in size than test tokens and $\sim 100'000$ tokens are sampled for logits computation. There is insufficient data from AG News to establish a meaningful public reference dataset, so we resort to ChatGPT to generate a synthetic one.

| Datasets | Heterogeneity | with the most task overlap | with the least task overlap |
|----------|---------------|----------------------------|------------------------------|
| AG News | Low | 0.39-0.41 | 0.31-0.32 |
| | High | 0.42-0.44 | 0.28-0.33 |
| Multi-Wiki | Low | 0.45-0.46 | 0.19-0.24 |
| | High | 0.45-0.47 | 0.19-0.23 |
| Codes-Wiki | Low | 0.40-0.41 | 0.39-0.42 |
| | High | 0.41-0.42 | 0.12-0.13 |

Table 3: Jaccard Index values for various datasets, indicating the range of token overlap among clients in different configurations.

3. Codes-Wikipedia (Eng): The first category is Java code from GitHub (HuggingFace) and the second category is English Wikipedia text (Wikimedia, 2022).

The corresponding distributed datasets are presented in Table 2. Note that AG News and Codes-Wikipedia simulate different topic distributions across users, while Multilingual Wikipedia simulates different language usage across users. It is worth noting that even though the topics/languages are different, they can nevertheless share the same tokens in their vocabularies. The portion of overlapped tokens is indeed not small, as measured by the Jaccard Index in Table 3. Jaccard Index for multi-sets, is calculated as $J(\mathcal{D}_i, \mathcal{D}_j) = \frac{|\mathcal{D}_i \cap \mathcal{D}_j|}{|\mathcal{D}_i| + |\mathcal{D}_j|} \in [0, 0.5]$. A high Jaccard Index indicates that clients' data share a substantial number of the same tokens. Although this measure doesn't account for the context information, it provides insight into the likelihood of finding collaborators. In all our experiments, the shared public dataset $X_S$ is sampled equally from each local distribution to ensure that the logit predictions are meaningful. In practice, such $X_S$ can be chosen as any publicly available dataset from the same domain, or private data with data sanitization procedures enforced on top to secure user privacy.

### 4.1.3 Baselines

Our chosen baseline methods are: 1) Local Fine-Tuning, where there is no communication between users, 2) FedAvg (McMahan et al., 2017), where model updates are aggregated to obtain a global model for each user, and 3) FedAvg + Local Fine-Tuning (Jiang et al., 2023), where we spare 10% of training data for local fine-tuning after FedAvg training. We offer an extended table where we vary the amount of training data used for fine-tuning for completeness in Table 9 in Appendix. We further incorporate a strong baseline – *oracle*. The determination of *oracle* collaboration weights relies on the similarity of underlying data source distributions, which is typically *unknown* in practice. We give an example of how the *oracle* weights are determined: if user 1 has 1/4 German texts and 3/4 French texts, and user 2 has 1/4 German texts and 3/4 Italian texts, the oracle weights would be determined by the dot product of $[1/4, 3/4, 0]$ and $[1/4, 0, 3/4]$. After obtaining the pairwise dot product matrix, row normalization is performed.

| Method | Multi-Wiki | | Codes-Wiki | |
| --- | --- | --- | --- | --- |
| | Low | High | Low | High |
| Local | 1 | 1 | 1 | 1 |
| FedAvg | 1.6 | 4.2 | 2.0 | 1.7 |
| Strategy 1 | 5.9 | 12.0 | 2.0 | 2.1 |
| Strategy 2 | 1.1 | 2.6 | 1.5 | NA |
| Strategy 3 | 2.9 | NA | 1.2 | 3.8 |

Table 4: Training time ($X$ times the needed training iterations as in the fully connected case) required for a ring topology to achieve the same perplexity level as a fully connected topology. NA indicates we did not reach the same perplexity after ten times the training iterations.

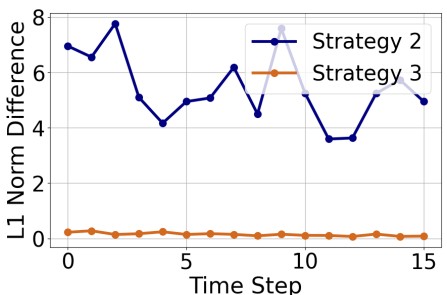

Figure 3: L1 norm of differences between $W$ at two consecutive steps for Strategy 2 and Strategy 3

## 4.2 Results

Our main findings are summarized in Table 5. Compared to baseline approaches—local training, FedAvg, and FedAvg + FT—our test performance and prediction-based aggregation methods consistently outperform them across all datasets and scenarios. We emphasize that in language modeling, achieving a balance between personalization and collaboration is not straightforward. Despite significant differences between categories, there are notable similarities in sentence structure and word choices, alongside domain-specific vocabulary. Thus, the preferences for collaboration or personalization vary across different datasets. For example, in the case of AG News, local training yields better results than FedAvg, whereas this is not true for Codes-Wiki. The collaboration-then-personalization approach (FedAvg + Local Fine-Tuning) fails to achieve this balance. In contrast, our methods successfully accomplish it.

Among our protocols, the predictions-based method exhibits the most superior test performance. Notably, the model-similarity-based aggregation method yields performance akin to FedAvg, where the trust weight is uniformly set to $1/N$ across all clients. Our methods can sometimes outperform *oracle* aggregation, suggesting that a *dynamic* collaborator selection protocol might be favored in different fine-tuning stages.

Ablation studies on the effect of LoRA ranks and local training set size can be found in Table 8 and Table 7 in Appendix A.3. Our prediction-based strategy consistently delivers superior performance.

**A note on communication topology:** While the paper demonstrates a fully connected communication topology, our protocols can support sparser communication patterns, as long as the graph represented by $W$ is strongly connected (i.e., every vertex is reachable from every other vertex), though convergence will be slower. We provide experimental results on the training time required for a ring topology to achieve the same perplexity level as a fully connected topology in Table 4. In high heterogeneity scenarios where each client is allocated a specific category, a ring topology results in cases where adjacent clients do not share any categories, leading to an *oracle* trust weight of 0 and hence the NA entries. Despite this, our algorithm can still learn, converging to a perplexity approximately one point higher than reported for the fully connected case.

## 4.3 Trust matrix

We compare our learned trust matrices with the *oracle* trust matrix in Figures 4 and 5. It is evident that Strategies 2 and 3 uncover similar collaboration patterns as suggested by the *oracle* matrix, whereas Strategy 1 falls short in this regard. Specifically, Strategy 1 assigns nearly identical trust weights to all other users, indicating nearly identical LoRA weights learned at convergence. This strange behavior is further investigated in Appendix A.4: even without communication, the LoRA weights across users are almost equally different,

| Heterogeneity | Method | Datasets | | |
|---|---|---|---|---|
| | | AG News | Multi-Wiki | Codes-Wiki |
| Low | Local | 30.17(0.17) | 40.00(0.33) | 19.57(0.23) |
| | FedAvg | 31.66(0.20) | 52.75(0.57) | 17.53(0.19) |
| | FedAvg +FT | 32.25(0.20) | 48.02(0.25) | 20.17(0.33) |
| | Strategy 1 | 31.43(0.35) | 45.59(0.66) | 17.57(0.21) |
| | Strategy 2 | **29.75(0.23)** | 36.93(0.17) | 17.61(0.40) |
| | Strategy 3 | 29.81(0.13)$^\star$ | **36.70(0.23)** | **17.35(0.18)** |
| | *Oracle* | *29.56(0.21)* | *39.55(0.19)* | *17.42(0.19)* |
| High | Local | 28.67(0.13) | 40.24(0.25) | 17.56(0.08) |
| | FedAvg | 32.08(0.13) | 53.23(0.51) | 16.68(0.06) |
| | FedAvg +FT | 33.74(0.17) | 48.07(0.18) | 18.43(0.18) |
| | Strategy 1 | 31.93(0.86) | 49.34(2.46) | 16.84(0.05) |
| | Strategy 2 | **28.29(0.06)** | 37.20(0.20) | 16.22(0.17) |
| | Strategy 3 | 28.72(0.27)$^\star$ | **36.92(0.16)** | **16.23(0.12)** |
| | *Oracle* | *28.08(0.11)* | *35.96(0.24)* | *16.20(0.05)* |

Table 5: Test perplexities (standard deviation) of our proposed strategies and baseline methods (lower is better). Strategy 1: model-similarity-based; Strategy 2: validation-performance-based, Strategy 3: prediction-similarity-based. $\star$ denotes that the used public dataset is a synthetic dataset we constructed using ChatGPT.

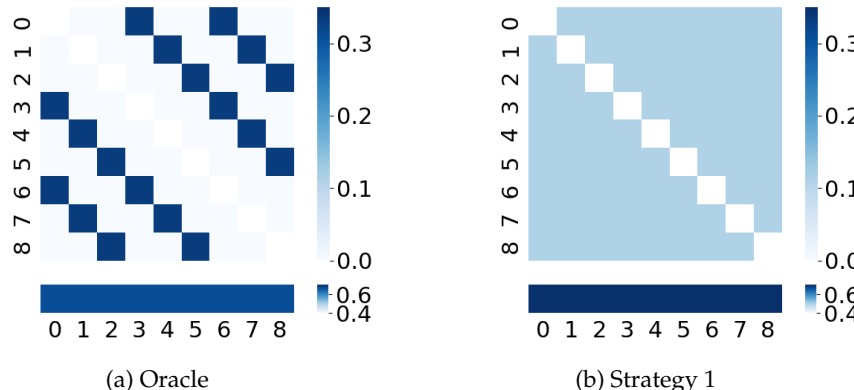

(a) Oracle              (b) Strategy 1

Figure 4: Oracle trust matrix versus learned trust matrix using Strategy 1 when users are allocated with Multilingual Wikipedia datasets. The diagonal entries are masked out and the trust is measured when the training ends.

suggesting that *LoRA models are not informative in collaborator selection when cosine similarity is used as the distance metric*.

The success of Strategies 2 and 3 underscores the advantage of utilizing predictions to identify collaborators. It is interesting how the distinctions between different users' models become more pronounced after a forward pass. Such a phenomenon was not previously observed in the vision domain. We further noticed that trust matrices in Strategy 3 are more stable across communication rounds compared to Strategy 2, with fewer abrupt changes in values. As shown in Figure 3, the $l_1$ norm of the differences between $W$s at consecutive steps fluctuates significantly for Strategy 2. It is speculated that the more stable trust matrix in Strategy 3 contributes to stabilizing the learning process, leading to better results.

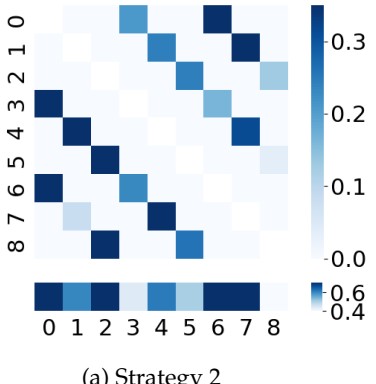
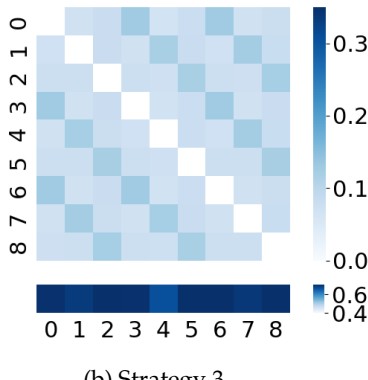

(a) Strategy 2          (b) Strategy 3

Figure 5: Learned trust matrix using Strategies 2 and 3 when users are allocated with Multilingual Wikipedia datasets. The diagonal entries are masked out and the trust is measured when the training ends.

## 5 Conclusions

Drawing from insights in conventional peer-to-peer collaborative learning approaches, we propose 3 collaboration protocols tailored for jointly on-device LLM fine-tuning. To our best knowledge, this is the first work demonstrating that collaboration can be leveraged to improve personalization performance in language modeling. Our methods can efficiently address challenges encountered in on-device personalized LLMs, such as limited availability of local data and data heterogeneity stemming from user characteristics.

We show that uniform collaboration as in federated averaging is not sufficient, but that dynamic collaborator selection strategies enable better and personalized models. Remarkably, for collaborator selection, *predictions are more informative than model weights*. Our Strategy 3, the prediction-similarity-based protocol, demonstrates promising practical applications for on-device deployment. It achieves superior performance while managing to maintain reasonable communication and computation costs.

## 6 Discussions and Future Work

In this study, we examined resource-constrained scenarios of collaborative LLM fine-tuning. Our collaborator selection scheme successfully addresses realistic data heterogeneity across users. However, heterogeneities can arise in different aspects, such as model and resource diversities. When users possess different capacities, enabling efficient collaboration remains an important challenge to address. Cho et al. (2024) have developed collaboration protocols tailored for circumstances wherein users exhibit varying levels of support for LoRA ranks. When users are equipped with distinct numbers of LoRA modules corresponding to their capacities, can we still harness the advantages of collaborative training?

Furthermore, within the scope of this research, particular emphasis is placed on unsupervised fine-tuning. The prospect of supervised fine-tuning, i.e. instruction tuning, is worth further investigation. An intriguing question arises: Can collaboration among users enhance their ability to generalize and improve performance on downstream tasks?

Our collaborator selection protocols are centered around *trust*. While the efficacy of trust in identifying collaborators has been thoroughly explored, there is potential for further investigation into its role in identifying adversarial users, which is one of the major challenges in federated LLMs (Chen et al., 2023). We finally note that our scheme can be extended to the case of partial participation of users and asynchronous updates, as the trust matrix could be updated less frequently than in every iteration, and decentralized averaging algorithms can tolerate asynchronous updates (Even et al., 2024).

**Acknowledgments**

We thank Soumajit Majumder and Bettina Messmer for their helpful review and all the anonymous reviewers for their constructive feedback. We acknowledge funding from Huawei Cloud Intelligent Cloud Technologies Initiative, Google Research Collaborations, and from the Swiss National Science Foundation (SNSF) grant number 200020_200342.

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

# A  Appendix

## A.1  Communication cost experiments

Tables 6 is a more detailed analysis of the communication costs within each communication round. In this specific case, communicating logits is the most costly. However, the cost can be largely reduced (to 0.014% of the original size) via sparse encoding techniques.

Note that different communicated units scale with different quantities. When the model size is larger, for example in billions, models and model updates can be larger than logits. In this case, logits can even emerge as a preferable communication unit over weights or gradients without compression.

| Communicated Elements | Communication Costs | Measured Costs (MB) |
|---|---|---|
| LoRA weights ($\boldsymbol{\theta}_i$) | $\mathcal{O}(Ldk)$ | 2.359 |
| LoRA model updates ($\Delta\boldsymbol{\theta}_i$) | $\mathcal{O}(Ldk)$ | 2.359 |
| Logits ($f_{\boldsymbol{\theta}_i}(\boldsymbol{X}_S)$) | $\mathcal{O}(V)$ | 40.206 |
| Sparse Top-K Logits ($f_{\boldsymbol{\theta}_i}(\boldsymbol{X}_S)_{COO}$) | $\mathcal{O}(K)$ | 0.560 |

Table 6: A comparison of communication complexity across the different shared units. $L$ denotes the number of layers, $d$ denotes the embedding size, $k$ is the LoRA rank, $V$ denotes the vocabulary size and $K$ denotes the number of logits kept for communication. *COO* denotes the sparse encoding used, *Coordinate list*.

## A.2  Hyper-parameters details

The choice of our hyper-parameters is influenced by resource constraints, primarily due to memory limitations (training conducted on a single NVIDIA A100 GPU with 40GB memory) and time constraints, as we needed to fine-tune up to 9 large language models (LLMs). Through empirical observation, we found that a batch size of 50 with 4 accumulation steps and a context length of 512 performed well within memory constraints. Additionally, while increasing the LoRA rank slightly improved test perplexity, the associated increase in training time rendered it uninteresting for our experiments. Therefore, we opted for LoRA rank 4. Three hyper-parameters remain to be tuned: learning rate, LoRA alpha, and LoRA dropout. In Figure 6, the best perplexity was achieved with a learning rate of 0.002, LoRA dropout of 0.1, and LoRA alpha of 32.

## A.3  Additional Ablations

### A.3.1  Impact of local dataset size

We increase the number of training tokens by 100 times, resulting in 84 million training tokens per client in the multilingual setup. The ranking of all methods remained consistent with what we report in Table 5.

| Method | Local | FedAvg | Strategy 1 | Strategy 2 | Strategy 3 | Oracle |
|---|---|---|---|---|---|---|
| val ppl | 37.80(0.15) | 56.69(0.35) | 48.47(2.55) | 36.17(0.30) | **35.63(0.15)** | 35.59 (0.16) |

Table 7: Comparison of methods on Multi-Wiki dataset

### A.3.2  Impact of LoRA ranks

In the main body, we restrict the LoRA rank to 4 to maintain a low computational budget. We conduct an ablation study on the LoRA ranks and present the results in Table 8. The rankings of the methods are consistent with our previous findings, with Strategy 3 being the best method. Additionally, as expected, a higher LoRA rank results in lower test perplexity.

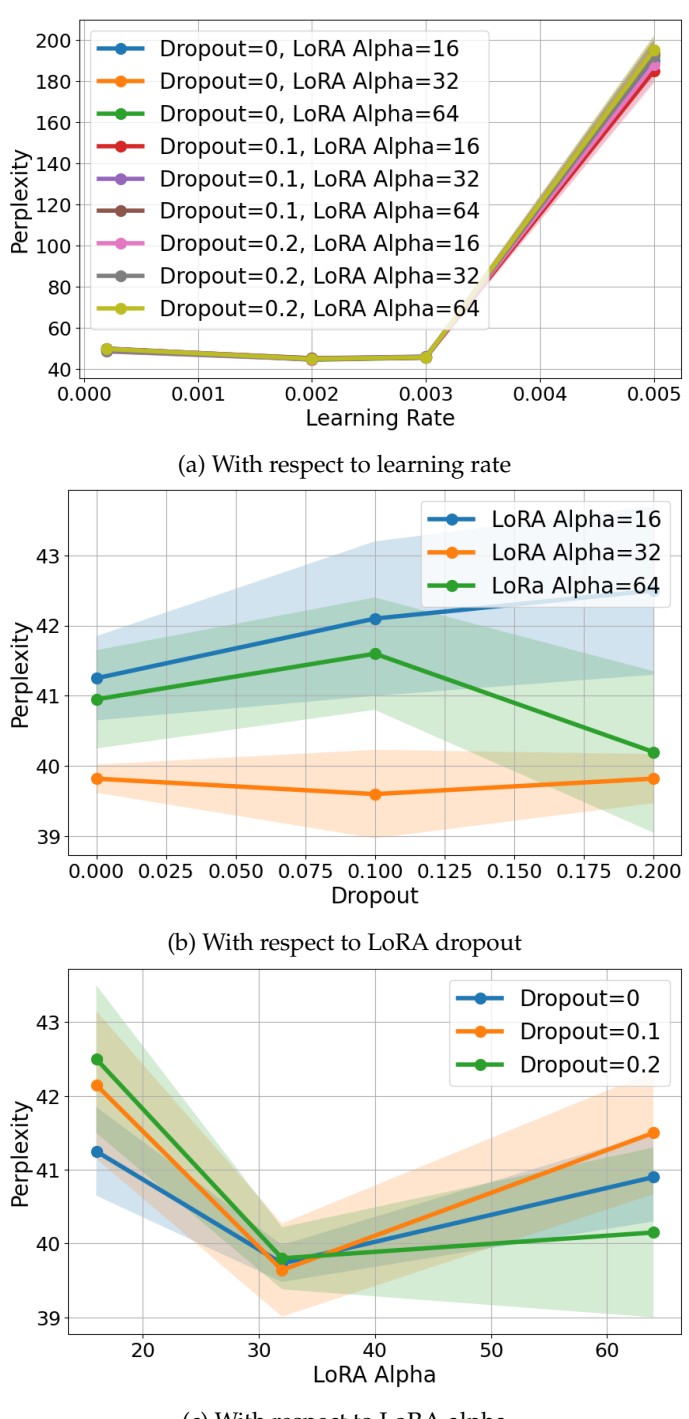

(a) With respect to learning rate

(b) With respect to LoRA dropout

(c) With respect to LoRA alpha

Figure 6: Hyperparameter tuning on learning rate, dropout, and LoRA alpha. LoRA dropout and LoRA alpha have a very small impact on the perplexity, the most important hyperparameter is the learning rate.

### A.3.3 FedAvg with Local Fine-Tuning

To understand better the relationship between the performance of FedAvg + Local Fine-Tuning and the relative size of the local fine-tuning dataset, we spare different percentages

| LoRA Ranks | 4 | 8 | 16 | 32 |
|---|---|---|---|---|
| Local | 40.24(0.25) | 33.15(0.15) | 29.14(0.11) | 27.41(0.11) |
| FedAvg | 53.23(0.51) | 48.84(0.34) | 42.68(0.06) | 40.12(0.25) |
| Strategy 1 | 49.34(2.46) | 48.37(1.26) | 40.50(0.11) | 34.65(5.73) |
| Strategy 2 | 37.20(0.20) | 30.41(0.13) | 26.36(0.12) | 24.37(0.20) |
| Strategy 3 | **36.92(0.12)** | **30.25(0.19)** | **26.30(0.11)** | **24.39(0.02)** |
| Oracle | 35.96(0.24) | 30.06(0.16) | 26.16(0.07) | 24.43(0.10) |

Table 8: Comparison of different methods with various LoRA ranks

of local data for fine-tuning purposes, and the results are presented in Table 9. With more local fine-tuning data, the performance gets closer to Local training without collaboration.

| Heterogeneity | Method | Datasets | | |
|---|---|---|---|---|
| | | AG News | Multi-Wiki | Codes-Wiki |
| Low | Local | 30.17(0.17) | 40.00(0.33) | 19.57(0.23) |
| | FedAvg | 31.66(0.20) | 52.75(0.57) | 17.53(0.19) |
| | FedAvg +FT (10%) | 32.25(0.20) | 48.02(0.25) | 20.17(0.33) |
| | FedAvg +FT (25%) | 31.29(0.15) | 43.74(0.24) | 20.10(0.25) |
| | FedAvg +FT (50%) | 30.03(0.23) | 39.79(0.21) | 19.59(0.26) |
| | Strategy 1 | 31.43(0.35) | 45.59(0.66) | 17.57(0.21) |
| | Strategy 2 | **29.75(0.23)** | 36.93(0.17) | 17.61(0.40) |
| | Strategy 3 | 29.81(0.13)$^\star$ | **36.70(0.23)** | **17.35(0.18)** |
| | *Oracle* | *29.56(0.21)* | *39.55(0.19)* | *17.42(0.19)* |
| High | Local | 28.67(0.13) | 40.24(0.25) | 17.56(0.08) |
| | FedAvg | 32.08(0.13) | 53.23(0.51) | 16.68(0.06) |
| | FedAvg +FT (10%) | 33.74(0.17) | 48.07(0.18) | 18.43(0.18) |
| | FedAvg +FT (25%) | 32.11(0.16) | 43.68(0.13) | 18.15(0.11) |
| | FedAvg +FT (50%) | 29.84(0.09) | 40.14(0.19) | 17.88(0.16) |
| | Strategy 1 | 31.93(0.86) | 49.34(2.46) | 16.84(0.05) |
| | Strategy 2 | **28.29(0.06)** | 37.20(0.20) | 16.22(0.17) |
| | Strategy 3 | 28.72(0.27)$^\star$ | **36.92(0.16)** | **16.23(0.12)** |
| | *Oracle* | *28.08(0.11)* | *35.96(0.24)* | *16.20(0.05)* |

Table 9: (Extended) Test perplexities (standard deviation) of our proposed strategies and baseline methods (lower is better). Strategy 1: model-similarity-based; Strategy 2: validation-performance-based, Strategy 3: prediction-similarity-based. $\star$ denotes that the used public dataset is a synthetic dataset we constructed using ChatGPT.

## A.4 Further analysis of model-similarity-based trust

We offer a comprehensive analysis to elucidate why Strategy 1 underperforms the other two strategies. The convergent model-similarity-based trust matrix resembling a uniform matrix is noteworthy. This occurrence is peculiar, as we would anticipate heterogeneity due to the varying distributions of users' data.

### A.4.1 Strategy 1 trust matrix converges to uniform even without communication

We let each user perform local fine-tuning for 500 iterations ($\sim$ 50 epochs). Figure 7 illustrates learned trust matrices using Strategy 1 in different global rounds. Note that here,

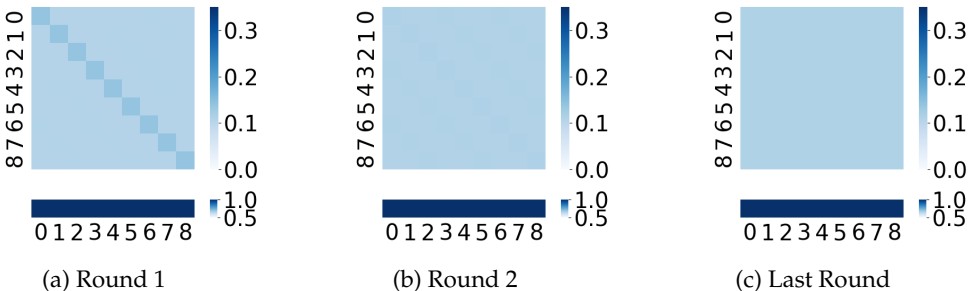

(a) Round 1           (b) Round 2           (c) Last Round

Figure 7: Learned trust matrix using Strategy 1 at different global rounds, without communication, when users are allocated with Multilingual Wikipedia datasets.

we use one global round to represent every 25 iterations, *without* any actual communication between users. Figure 8 shows the learned trust matrices at convergence using the three different strategies.

Even without communication, the learned trust matrix using Strategy 1 becomes uniform. This indicates that the users learned almost identical LoRA weights.

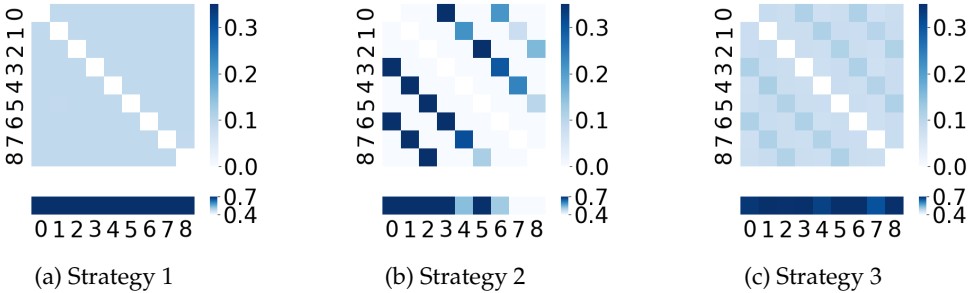

(a) Strategy 1           (b) Strategy 2           (c) Strategy 3

Figure 8: Learned trust matrix using Strategies 1, 2, and 3 when users are allocated with Multilingual Wikipedia datasets. No communication and aggregation are performed.

### A.4.2 Strategy 1 learns the collaboration pattern, but at a tiny scale

We dive into this bizarre observation. In Figure 9, we plot out the exact same trust matrices as in Figure 7, but present with a much smaller scale. This visualization indicates that Strategy 1 indeed can identify users who are more helpful to collaborate with, but the distinction between more and less helpful collaborators is not pronounced.

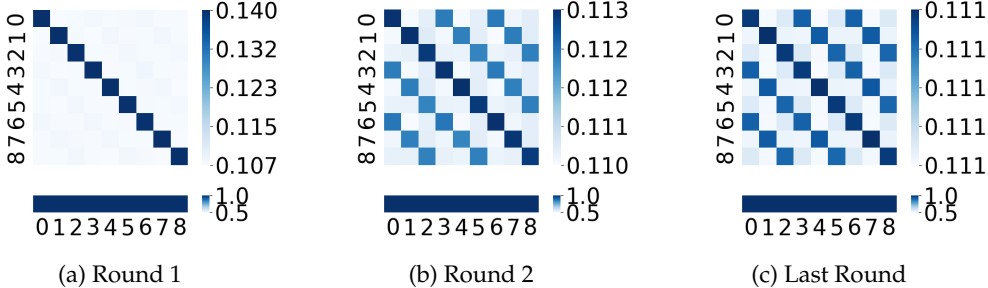

(a) Round 1           (b) Round 2           (c) Last Round

Figure 9: Learned trust matrix using Strategy 1 at different global rounds, without communication, when users are allocated with Multilingual Wikipedia datasets. Displayed with a smaller scale.

