# OpenReview forum: "Personalized Collaborative Fine-Tuning for On-Device Large Language Models"
_colmweb.org/COLM/2024/Conference — COLM_

### Official Review · Reviewer_nijS · 2024-05-10

**Rating:** 7
**Confidence:** 4
**Ethics Flag:** 1

**Summary:**

This paper proposes an approach to fine-tune personalized language models in a collaborative setting. Specifically, the idea is to derive a set of LoRA weights for each user by aggregation across other users who are similar to this user. The paper presents 3 strategies for aggregation based on a) similarity of weights, b) similarity of predictions based on user specific validation sets, i.e. how well LORA weight of user j performs on validation set of user i, c) similarity of predictions based on a shared dataset. The paper finds prediction based similarity works best and furthermore using a shared dataset works somewhat better.

Strengths:
* Presents an effective strategy for developing personalized language models while still benefiting from other users in a collaborative setting
* Demonstrates that a weight aggregation strategy based on predictions outperforms one based on model-similarity.

Weaknesses:
* In a realistic scenario, users may be wary of sharing information with other users. Therefore, we may want to add some level of noise to a user's prediction or LoRA weights before using those weights in an aggregation setting. The paper does not discuss this privacy angle.
* The paper does not report an ablation wrt the amount of fine-tuning data and how that can affect the performance of the 3 strategies.

**Questions To Authors:**

* Why is strategy 3 not reported for AG News?
* In Figure 1, should  \delta \theta_2, s_2 be \delta \theta_3, s3?
* In Sec 3.2, "Essentially, the weight of edge ( i , j ) should denote to what extent can user j ’s gradients help to facilitate user j ’s learning progress" should this be  "user i's gradients help to facilitate user j's ..." ?

**Reasons To Accept:**

* The paper presents an effective strategy to train personalized language models in a collaborative setting. Such an approach seems worthwhile given that lot of meaningful data is in user specific private repositories and unavailable for LLM training
* The paper compares multiple strategies for aggregating user weights and demonstrates that a prediction based strategy works better than one based on model-similarity, a good finding

**Reasons To Reject:**

* In realistic scenarios, users are concerned about privacy. Thus, it would be important to add noise to the user's weights or predictions before incorporating them in an aggregation setting. The paper does not discuss this aspect.
* In a collaborative setting, users exhibit heterogeneity in terms of usage I.e. in terms of amount of fine-tuning data available. The paper does not discuss this dimension and how it may impact performance of the proposed approach.

---

> ### Author Rebuttal · Authors · 2024-05-31
>
> Thanks for your positive feedback.
>
> Regarding privacy concerns, while the paper didn't extensively address them, we ensured a degree of privacy by keeping data on local devices. However, we agree that discussing privacy implications further would enhance practical value. Our primary focus is demonstrating that collaboration can enhance personalization performance in language modeling, which is a significant contribution. We also conducted experiments in realistic language heterogeneity scenarios to empirically validate our strategies.
>
> Regarding the ablation on the size of fine-tuning data, we conducted supplementary experiments, increasing the training tokens by 100 times, resulting in 84 million training tokens per client in the multilingual setup. The ranking of all methods remained consistent with our paper's reported results.
>
>
> | Local | FedAvg |Strategy 1 | Strategy 2 | Strategy 3 | Theoretical |
> | ----------- | ----------- |----------- |----------- |----------- |----------- |
> | 37.80 (0.15) | 56.69 (0.35)|48.47 (2.55) |36.17 (0.30)|35.63 (0.15)|35.59 (0.16)|
>
>
> With regard to your raised questions, we answer them as follows:
>
> - The AG News dataset is very small, and we do not have enough data to construct a shared dataset across the clients, which is why we left those entries empty. However, we explored other possibilities for obtaining a publicly shared dataset and generated some synthetic news articles using ChatGPT. With this synthetic shared data, we obtained the following results: our strategy 3 performs on par with strategy 2, and both outperform all other baseline methods by a large margin. This further illustrates the practical potential of strategy 3, as one can easily create such synthetic datasets.
>
>
> | Method | Local | FedAvg | Strategy 1 | Strategy 2 | Strategy 3 | Theoretical |
> | ----------- | ----------- | ----------- | ----------- |----------- |----------- |----------- |
> Low heterogeneity | 30.17 (0.17) | 31.66 (0.20)|31.43 (0.35) |29.75 (0.23)|29.81 (0.13) |29.56 (0.21)
> High heterogeneity | 28.67 (0.13) | 32.08 (0.13)|31.93 (0.86) |28.29 (0.06)|28.72 (0.27) |28.08 (0.11)
>
>
> - Yes you are right about the mistake in Figure 1. We will make sure this is corrected in future iterations.
>
> - Thanks for pointing this typo out, it should be “ the weight of edge (i, j) should denote to what extent can user j ’s gradients help to facilitate user i’s learning progress.”

---

> > ### Comment · Reviewer_nijS · 2024-06-03
> >
> > Thanks for the clarifications.

---

### Official Review · Reviewer_jbCH · 2024-05-10

**Rating:** 5
**Confidence:** 4
**Ethics Flag:** 1

**Summary:**

This paper investigates a personalized collaborative fine-tuning of LLM on-device, proposing three trust-weighted gradient aggregation strategies to deal with local data heterogeneity and scarcity problems. The significance of the work lies in the current landscape where data privacy, device-centric ML and federated learning are growing in importance. By pushing forward with collaborative learning directly on devices through parameter efficient fine tuning, this approach has the potential to open up new avenues for more personalized, secured and efficient applications. The paper succinctly explains three strategies, conducting limited experiments in unsupervised fine-tuning tasks, comparing them with one SOTA federated learning method. Additionally, it provides supplementary experiments aimed at interpreting the underwhelming performance of the first strategy. However, the lack of theoretical explanation or analysis for this fact and the other two strategies is notable. The paper is clear and easy to understand, although some ambiguity may arise from certain figures and letter symbols. Nevertheless, the three strategies lack significant novelty compared to the referenced work and do not introduce substantial modifications or improvements.

**Reasons To Accept:**

Developing device-centric collaborative parameter efficient fine-tuning methods to address data privacy concerns and heterogeneity challenge holds significant promise.

**Reasons To Reject:**

1. The paper lacks sufficient experiments with multitask and larger datasets, and controlled experiments to demonstrate the methods’ generalization performance claimed by the authors. The three strategies were only tested in unsupervised fine-tuning task (focus on diverse topic, no considering other variation like writing styles, vocabulary choices, grammar structures, etc.), with perplexity as the sole evaluation metric. The dataset size is relatively small (approximately 1600 samples based on the provided token count and context length). Additionally, the comparison with the sole federated learning algorithm, FedAvg, and the fixed rank of the LoRA adapter (set to a small value of 4) limit the persuasiveness of the paper.
2.  Some figure and letter symbols may cause ambiguity. From the Algorithm, each user should broadcast their LoRA weight update to all others agents, while in Figure 1 just user 1, 3, 4 broadcast their updates to user 2. What does the symbol si message refer to? (LoRA weight for user i?)
3. Lack of novelty. Three strategies do not introduce substantial modification compared to the corresponding three referenced works. According the the experiment results, these methods exhibit limited improvement compared to the baselines, even perform worse than the simple but strong baseline theoretical but consume much more computing power.
4. Lack of theoretically explanation to the underwhelming performance of Strategy 1 and that of Strategy 2 and 3 compared to the strong baseline. Some facile explanations or assumptions (like "This could be attributed to the high-dimensional, sparse nature, and small magnitudes of LoRA weights"), but lack of further experiment to verify them.
5. Lack of limitation discussion.

---

> ### Author Rebuttal · Authors · 2024-05-31
>
> 1. We acknowledge our test scenarios are limited due to computing resources. Regarding datasets, we argue our constructed scenarios reflect realistic language heterogeneity, encompassing variations in language usage, topic-specific wording, and data-source-specific wording.
>
> Additionally, as suggested, we tested our methods on a larger dataset (84 million training tokens per client, 100 times larger than in the paper, 1st Table below). We also compared our strategies against baseline methods across different LoRA ranks (2nd Table). Both show consistent trends as in the paper, confirming the validity of our methods. (S denotes strategy in tables)
>
> | Local | FedAvg |S1 | S2 | S3 | Theoretical |
> | ----------- | ----------- |----------- |----------- |----------- |----------- |
> | 37.80 (0.15) | 56.69 (0.35)|48.47 (2.55) |36.17 (0.30)|35.63 (0.15)|35.59 (0.16)|
>
> | LoRA Ranks      | 4 |8 | 16 | 32 |
> | ----------- | ----------- |----------- |----------- |----------- |
> | Local | 40.24 (0.25) |33.15 (0.15)| 29.14 (0.11)| 27.41 (0.11)|
> | FedAvg   | 53.23 (0.51) | 48.84 (0.34) |42.68 (0.06) |40.12 (0.25)|
> | S1 | 49.34 (2.46)| 48.37 (1.26) | 40.50 (0.11) |34.65 (5.73)|
> | S2 | 37.20 (0.20) |30.41 (0.13) |26.36 (0.12) |24.37 (0.20)|
> | S3  | 36.92 (0.12) |30.25 (0.19) |26.3 (0.11) |24.39 (0.02)|
> | Theoretical | 35.96 (0.24) |30.06 (0.16)| 26.16 (0.07) |24.43 (0.10)|
>
> Clarifications on symbol ambiguity: The algorithm assumes an all-to-all communication graph. While Figure 1 illustrates one client for readability, communication occurs in parallel for each client. Symbol $s_i$ denotes the message client i sends apart from $\Delta \theta_i$, which varies by strategy. Their meanings and their extra computation costs can be found in Table 1 in the paper.
>
> 2. Methodologically, we acknowledge limited novel contributions. However, this is the first work demonstrating collaboration leveraged to improve personalization performance (each client has a personalized state) in language modeling, which is a significant contribution.
> Note that the  _theoretical_ baseline is not available in practice, as it requires information on underlying data mixtures on the device. The theoretical baseline helps to demonstrate how well our method can perform even without knowing data distribution a priori.
>
> 3. Regarding why strategy 1 underperforms, please refer to Figure 8 in the appendix: In strategy 1, the trust matrix converges to a uniform matrix during training, resembling FedAvg's performance.

---

> > ### Comment · Reviewer_jbCH · 2024-06-05
> >
> > Thanks for providing more details. I would like to my rating the same.

---

### Official Review · Reviewer_E7HE · 2024-05-11

**Rating:** 5
**Confidence:** 4
**Ethics Flag:** 1

**Summary:**

This paper introduces a method for on-device collaborative fine-tuning of large language models using Low-Rank Adaptation (LoRA). The approach is distinct for its minimal communication overhead and integration of three innovative trust-weighted gradient aggregation schemes: weight similarity-based, prediction similarity-based, and validation performance-based. Through their experiments, the author shows that the aggregation trust matrix guided by prediction information (strategy 2 and 3) yields better performance than the one based on similarities.

**Reasons To Accept:**

- The three strategies mentioned in this paper are easy to implement and do not require heavy hyperparameter search.
-  The experiments are robust, covering different data heterogeneities and showing clear benefits of the proposed methods over traditional FedAvg and local fine-tuning.
- The paper provides the analysis on the superiority of prediction-based aggregation methods (strategy 2 and 3), which helps in understanding why these methods might be more effective.

**Reasons To Reject:**

- While the paper demonstrates that strategy 3 consistently outperforms strategy 2, there is no detailed analysis explaining why this is the case. Since both strategies use prediction-guided trust matrices, additional insights into their differences would strengthen the conclusions.
- The methods may not scale well with a large number of clients, which is a significant limitation given the growing scale of data and model deployments.
- The primary contribution seems to be the application of existing techniques (from Zhang 2021 and Fan 2023) to LoRA, which makes the contribution of the paper a bit incremental.
- To enrich the comparative analysis and foster a deeper discussion, it is suggested that the manuscript incorporate several relevant on-device federated studies such as [1,2,3,4,5].


[1] "Federated Full-Parameter Tuning of Billion-Sized Language Models with Communication Cost under 18 Kilobytes" (arXiv'23, ICML'24)

[2] "FwdLLM: Efficient FedLLM using Forward Gradient" (arXiv'23)

[3] "Federated Fine-tuning of Large Language Models under Heterogeneous Language Tasks and Client Resources" (arXiv'24)

[4] "Can Public Large Language Models Help Private Cross-device Federated Learning?" (arXiv'23, NAACL'24)

[5] "FS-Real: Towards Real-World Cross-Device Federated Learning" (KDD'23)

---

> ### Author Rebuttal · Authors · 2024-05-31
>
> Thanks for your helpful feedback.
>
> Regarding the differences between strategies 2 and 3, we identified three major points:
> 1). Strategy 2 learns a sparser pattern than strategy 3, with many entries in the trust matrix being close to 0, unlike in strategy 3.  See Figure 4 of the paper. 2). We observed that the trust matrix in strategy 3 is more stable across communication rounds compared to strategy 2, with fewer abrupt changes in values. 3). The trust matrices from strategy 3 have higher diagonal values than those from strategy 2.
>
> We cannot definitively explain why strategy 2 underperforms, but we speculate that the more stable trust matrix and higher self-trust in strategy 3 contribute to stabilizing the learning process, leading to better results. We plan to include more visual evidence on the evolution of learned trust matrices in future iterations.
>
> Yes, we acknowledge that the method cannot scale up very well to a large number of clients. Being difficult to scale up is a general problem in all collaborator-selection type personalized methods, including ours. However, we emphasize that in this project, we target stateful clients (each client has a personalized model state) instead of stateless clients as in cross-device setup[1].
>
> Regarding the incremental contribution comment, methodologically, we acknowledge that we are not making novel contributions. However, to the best of our knowledge, this is the first work demonstrating that collaboration can be leveraged to improve personalization performance in language modeling.  By personalization, we mean that each client has a personalized state. We consider this a significant contribution. In language modeling, the balance between personalization and collaboration is not straightforward. Despite the drastic differences between categories, there are notable similarities in sentence structure and word choices, alongside domain-specific word choices. We would also like to draw your attention to Table 4, which shows that in different scenarios, the performances of FedAvg and Local methods vary significantly, and our collaborator-selection-based method can give consistently good performances.
>
> Thanks for the related work suggestions. We will make sure to include them in future iterations.
>
> [1] Kairouz et al. Advances and Open Problems in Federated Learning. Foundations and Trends in Machine Learning Vol 4 Issue 1.

---

### Official Review · Reviewer_pToR · 2024-05-17

**Rating:** 6
**Confidence:** 4
**Ethics Flag:** 1

**Summary:**

This paper discusses weighted aggregation methods for the LoRA module in decentralized learning. When freezing base networks, the LoRA modules are weighted by “trust” calculation before aggregation. The “trust” weights are computed by weights similarity, validation set, and prediction similarity.



========== after rebuttal and discussion ===============
I thank the authors for the response and raised the score to 6.

Why not lower:
I appreciate the clarification and extra work during rebuttal, and acknowledge the contribution of discussing collaboration and personalization of LoRA training of LLM in decentralized training. This is timely and relevant to COLM community.

Why not higher:
As also mentioned by other reviewers, the strategies discussed are not entirely new, and somewhat ad-hoc to me. The motivation of "realistic" heterogeneity makes sense, but I am still a little skeptical about the setting and baseline choices in the draft.

Finally, I would strongly encourage the authors to take the discussion into consideration, and more carefully adjust the technical terms and relationship with the literature.

**Questions To Authors:**

Could the authors explain how the "theoretical" matrix is computed in experiments?

**Reasons To Accept:**

The topic is relevant to the COLM conference.

**Reasons To Reject:**

I may have missed something, but in my humble opinion, this paper does not meet the bar of top conference.

1) The motivation is not quite clear to me. The method seems to aim for cross-device federated learning, but it seems to also assume full participation of clients every round instead of a small subset of clients. Moreover, it seems to assume a fully connected network topology among all clients, i.e., every communication is a full reduce step. It is challenging for me to understand this setting compared to the common setting, e.g., [Advances and Open Problems in Federated Learning](https://arxiv.org/abs/1912.04977) Section 1.

2) I somewhat feel strongly and disagree with users' usage of "trust". The weights for aggregation seems to merely depend on some kind of similarity. "Trust" is often a terminology for privacy and security, and it is hard for me to understand what the connection with "trust" is. On the other hand, the authors do not seem to properly discuss the privacy implication of the settings and the algorithms.

3) The proposed weighted aggregation methods are somewhat ad-hoc and lacking strong supports. The proposed method did not compare with any baseline method. There are many personalization methods in FL, and they can be applied to the LoRA training setting, e.g., [Motley: Benchmarking Heterogeneity and Personalization in Federated Learning](https://arxiv.org/abs/2206.09262) and methods therein.

---

> ### Author Rebuttal · Authors · 2024-05-30
>
> Thanks for your comments and feedback.
>
> Regarding the motivation, we are not targeting cross-device federated learning; our goal is to achieve a **personalized state** for each client, resembling more of a cross-silo setup, which requires stateful clients. This paper focuses on understanding personalization and collaboration within the context of privacy-preserving language models. Our experimental setup is similar to [1]. Although we considered a fully connected setup here, partial participation per communication round is orthogonal to our scheme and can be integrated. However, investigating the effects of partial participation in detail is beyond the scope of this paper.
>
> Regarding the term "trust," you raise a valid point. In this project, our use of the term is motivated by social science literature: clients can be seen as members of social groups who make decisions (personalized models) through interaction with other members. This interaction, performed via collaborator selection, depends on the trust between group members, with similarity serving merely as a metric. We will clarify this in future versions.
>
> We want to clarify that we are not providing state-of-the-art personalized federated learning algorithms. While many methods are applicable, this is the first work showing that collaboration improves stateful personalization in language modeling. Moreover, we have baseline methods including: 1) FedAvg and local training, demonstrating the need for _both_ personalization and collaboration for better validation perplexity; 2) a theoretical baseline using a collaboration graph based on underlying data mixture similarities.
>
> The theoretical matrix assumes _known_ data mixtures for each client. For instance, if client 1 has 1/4 German and 3/4 French texts, and client 2 has 1/4 German and 3/4 Italian texts, the theoretical weights are the dot product of [1/4, 3/4, 0] and [1/4, 0, 3/4], followed by row normalization. While not necessarily the optimal trust matrix, it uses underlying mixture information (_unrealistic_ in practice), hence "theoretical". It is worth noting that our strategy can even outperform this baseline, emphasizing the importance of a dynamic collaboration graph.
>
> Hope those clarifications clear your doubts. We welcome further discussions and would appreciate a score adjustment if we manage to address your raised concerns. Thank you.
>
> [1] Sun et al. Improving LoRA in Privacy-preserving Federated Learning. ICLR 2024

---

> > ### Comment · Reviewer_pToR · 2024-05-31
> >
> > Thanks for the response.
> >
> > 1. Could the authors clarify again about the main contribution of the paper? A few things I hope to understand more a) the fully connected topology seems to be a strong assumption; consider the empirical nature of this study, I am not convinced that extending to other topology and/or partial participation is easy for the reweighting strategies studied in this paper. b) related to a) there are many decentralized algorithms studied before, maybe I missed it, but I am not sure if they are discussed at all. For example [𝐷2: Decentralized Training over Decentralized Data] c) if the main focus is the reweighting strategies, it is also a problem that has been extensively studied in the literature, see section 3.2.4 in [A Field Guide to Federated Optimization] for papers before 2021, and more recent papers like [Revisiting Weighted Aggregation in Federated Learning with Neural Networks]. d) if the main focus in personalization, I am not convinced that a simple baseline of FedAvg + local finetuning is out of scope for comparison. Note that I did not ask for an extensive table like in [pFL-Bench: A Comprehensive Benchmark for
> > Personalized Federated Learning] which benchmarks 10+ methods, but IMHO, ignoring the entire literature seems a little disappointing. e) if this paper considers their contribution is to study a problem that has been largely overlooked before, please clarify the motivation related to question a) about topology and assumption. And carefully clarify their relationship with the literature.
> >
> > 2 I can accept the "trust" term if the authors can provide a few concrete reference that uses this term in this manner. They can be from the social science literature. Otherwise I would strongly encourage a replacement like "similarity" or something.
> >
> > 3 Following settings in existing papers is a great practice. But I also want to highlight that experiments should be designed to support the contributions of the paper. The focus of  [Sun et al. Improving LoRA in Privacy-preserving Federated Learning] is quite different from this paper, and hence I may need the authors to clarify why this setting is supporting their contributions. See 1 before.
> >
> > 4 I thank the author and acknowledge the explanation of "theoretical" in experiments. I would probably use a term like "oracle" if I were writing the paper.
> >
> > My main question is 1 about understanding the contribution of this paper in the literature.

---

> > > ### Author Response · Authors · 2024-06-03
> > > **Clarification of our contributions**
> > >
> > > Thanks for your quick response.
> > >
> > > 1. Regarding topology and partial participation, we understand that they should naturally be compatible with our current scheme. For instance, consider a specific client i. In rounds where client i is selected, we reweight all other selected clients connected via edges in that round. If no clients have a similar data distribution, our scheme assigns high self-trust and low trust towards others. Conversely, if there are clients with similar data distributions, our scheme assigns higher trust to these clients, allowing client i to make a better update in that round. A more sparse topology and low participation ratio will require more communication rounds to converge.
> > >
> > >
> > >  We agree FedAvg + Local FineTuning would be a valid baseline to compare to, but in our experimental setup, we only have a training set (where we train models) and validation set (where we report the test performance, it is more like a test set).  To our understanding, local fine-tuning requires some additional personalized data before evaluation, which we do not have. Therefore, we allocated a certain ratio (% in the parentheses) of the training data for fine-tuning and kept the rest for FedAvg training. The results are shown in the table below. With fine-tuning, the perplexity even went higher for some scenarios. We speculate that this may be due to the reduced amount of data available for FedAvg training and overfitting on personalized data.
> > >
> > >
> > > Method | AGNews Low | AGNews High | Multi-Wiki Low | Multi-Wiki High | Codes-Wiki Low | Codes-Wiki High |
> > > | ----------- | ----------- | ----------- | ----------- | ----------- |  ----------- |----------- |
> > > Local| 30.17 (0.17)| 28.67 (0.13) |40.00 (0.33) |40.24 (0.25) | 19.57 (0.23)|17.56(0.08)
> > > FedAvg | 31.66 (0.20)| 32.08 (0.13) | 52.75 (0.57) |53.23 (0.51) | 17.53 (0.19)| 16.68 (0.06)
> > > FedAvg + FT (10%) | 32.25 (0.20) | 33.74 (0.17)| 48.02 (0.25) | 48.07 (0.18)| 20.17 (0.33)| 18.43 (0.18)|
> > > FedAvg + FT (25%) | 31.29 (0.15)| 32.11 (0.16) |43.74 (0.24)|43.68 (0.13)| 20.10 (0.25) |18.15 (0.11)
> > > FedAvg + FT (50%) | 30.03 (0.23) | 29.84 (0.09) | 39.79 (0.21) | 40.14 (0.19) | 19.59 (0.26) | 17.88 (0.16)
> > > S1 |31.43 (0.35) |31.93 (0.86)|45.59 (0.66)|49.34 (2.46)|17.57 (0.21) |16.84 (0.05)
> > > S2 |29.75 (0.23)|28.29 (0.06)|36.93 (0.17)|37.20 (0.20) |17.61 (0.40)|16.22 (0.17)
> > > S3 | - |-|**36.70 (0.23)** |**36.92 (0.16)** |**17.35 (0.18)**|**16.23 (0.12)**|
> > >
> > > Please note that we did not claim to propose new decentralized personalization algorithms. In our opinion, the **main contribution** of this paper lies in two aspects: 1) we explore _realistic_ data heterogeneity among users in decentralized language modeling; and 2) we demonstrate that collaboration can be leveraged to improve personalization performance in language modeling. By personalization, we mean that each client has a personalized state. We hope the FedAvg + FT experiment illustrates that balancing personalization and collaboration is not straightforward in language modeling. Despite the drastic differences between categories, there are notable similarities in sentence structure and word choices, alongside domain-specific word choices, unlike typical classification tasks. Additionally, the preferences for collaboration or personalization vary across different datasets. For example, in the case of AG News, local training yields better results than FedAvg, whereas this is not true for Codes-Wiki. Our strategies can intelligently adapt to these different scenarios.
> > >
> > > 2. Yes, here are some references:
> > > - "Similarity measures could be useful to assess trust between users. For example, in opinion dynamics models, it has been proved that users tend to consider and trusting more other users with similar opinions to them” [1]
> > > - “The trust degree between individuals derived from opinion similarity at time t.[2]
> > >
> > > If we view clients as members of a social group, their model state or predictions on a shared dataset can thus be regarded as opinions.
> > >
> > > 3. We adopt the experimental setup as we focus on the same cross-silo scenario in the language domain. Please note that there are not many existing works that studied decentralized learning in language modeling. We think such an experimental setup demonstrates our purposes (see our clarification on contributions in point 1)
> > >
> > > 4. Yes that’s a good suggestion, we will adopt this term in future iterations.
> > >
> > >
> > > [1] Ureña et al. A review on trust propagation and opinion dynamics in social networks and group decision making frameworks
> > >
> > > [2] Zhang et al. Consensus reaching with trust evolution in social network group decision making

---

> > > > ### Comment · Reviewer_pToR · 2024-06-03
> > > >
> > > > I thank the authors for the efforts, and raised the score to 5.
> > > >
> > > > I am not fully convinced on my previous concern 1 about the contributions of this paper and position it in the literature, particularly a) justify the motivation of fully connected topology of decentralized learning b) justify the insights behind the three strategies studied in this draft and c) justify FedAvg/LocalTraining as the main baseline. I acknowledge c) has been partially addressed by the new results of FedAvg + FT.
> > > >
> > > > I acknowledge the contribution on heterogenous data and the motivation of combining collaboration and personalization. I would encourage the authors to improve the writing, and happy to further engage before discussion ends.

---

> ### Author Response · Authors · 2024-06-05
> **More clarifications**
>
> Thanks for your response and score adjustment. We will include your helpful comments in future iterations.
>
> We address your concerns as follows:
>
> - We opted for a fully connected topology because we are focusing on a cross-silo scenario with a small number of clients. This is the case for the standard federated learning setup, where the server can easily relay meta-data between all clients (see Table 2.1 in [4]). In the more general decentralized case where a different  connected topology is preferred, one can replace the current trust matrix W with a sparse version, where some entries are zero. Our setting thus includes both federated learning as well as arbitrary decentralized training. As long as the graph represented by W is strongly connected (i.e., every vertex is reachable from every other vertex), our algorithm should still function, though convergence will be slower.
>
> We provide experimental results on the training time (X times the needed training iterations as in the fully connected case) required for a ring topology to achieve the same perplexity level as a fully connected topology in the table below. Note that NA indicates we did not reach the same perplexity after ten times the training iterations. In high heterogeneity scenarios with a very sparse ring topology, each client is allocated a specific category. This can result in cases where adjacent clients in the ring do not share any categories, leading to an oracle trust weight of 0 and hence the NA entries. Despite this, our algorithm can still learn, converging to a perplexity approximately one point higher than what is reported for the fully connected case.
>
> Method | Multi-Wiki Low | Multi-Wiki High | Codes-Wiki Low | Codes-Wiki High |
> | ----------- | ----------- | ----------- | ----------- | ----------- |
> Local | 1 | 1 | 1| 1 |
> FedAvg | 1.6 | 4.2 | 2.0| 1.7 |
> S1 | 5.9 | 12.0 |2.0 | 2.1 |
> S2 | 1.1 | 2.6 | 1.5 | NA |
> S3 | 2.9 | NA |1.2 | 3.8 |
>
> 2. We summarize the insights as follows:
> - From strategy 1 underperforming the other two strategies, we conclude that predictions are more informative than model weights in identifying collaborators in language modeling.
> - Both as predictions-based collaborator selection methods, strategy 3 outperforms strategy 2. We further investigated the evolution of learned trust matrices. We found the trust matrices in strategy 3 are more stable across communication rounds compared to strategy 2, with fewer abrupt changes in values, and the trust matrices from strategy 3 have higher diagonal values than those from strategy 2. We cannot definitively explain why strategy 2 underperforms, but we speculate that the more stable trust matrix and higher self-trust in strategy 3 contribute to stabilizing the learning process, leading to better results.
>
> 3. As we mentioned in an earlier response, we are not proposing a SOTA method, instead we aim to understand how to balance personalization and collaboration. Moreover, as we focus on a server-less setup, the standard way is to use gossip averaging algorithm (which does collaborator selection), without getting a global model state involved [1,2,3]. That is also a reason why we did not compare personalization algorithms where a server is assumed.
>
> Thanks a lot and we welcome further discussions if things are unclear.
>
> [1] Sui et al. Find your friends: personalized federated learning with the right collaborators
>
> [2] Zhang et al. Personalized Federated Learning with First Order Model Optimization
>
> [3] Fan et al. Collaborative Learning via Prediction Consensus
>
> [4] Kairouz et al. Advances and Open Problems in Federated Learning

---

### Author Response · Authors · 2024-06-05
**Request for Participation in Discussion**

Dear Reviewers,

We kindly request that you review our response and join the discussion if you have not done so. As the discussion period is nearing its conclusion, we hope our efforts have adequately addressed your concerns. Thank you very much!

Best,
Submission 837 authors

---

### Decision · Program_Chairs · 2024-07-10

**Decision:**

Accept

**Comment:**

The paper discusses methods for collaborative / distributed fine tuning, and introduces several such "trust based" methods. Experiments compare their methods to federated learning and other known methods for aggregating ML models.
The problem is important and I find the paper to be well written.
Reviewers ranged from lukewarm to positive, and I recommend acceptance. Please address the comments by the reviewers.